# New Insight into Justicidin B Pathway and Production in *Linum austriacum*

**DOI:** 10.3390/ijms22052507

**Published:** 2021-03-02

**Authors:** Iride Mascheretti, Michela Alfieri, Massimiliano Lauria, Franca Locatelli, Roberto Consonni, Erica Cusano, Roméo A. Dougué Kentsop, Marina Laura, Gianluca Ottolina, Franco Faoro, Monica Mattana

**Affiliations:** 1Institute of Agricultural Biology and Biotechnology, National Research Council, Via Bassini 15, 20133 Milan, Italy; iride.mascheretti@ibba.cnr.it (I.M.); massimiliano.lauria@ibba.cnr.it (M.L.); franca.locatelli@ibba.cnr.it (F.L.); romeo.dougue@ibba.cnr.it (R.A.D.K.); 2Institute of Chemical Sciences and Technologies “Giulio Natta”, National Research Council, Via Mario Bianco 9, 20133 Milan, Italy; michela.alfieri@scitec.cnr.it (M.A.); roberto.consonni@scitec.cnr.it (R.C.); erica-cusano@hotmail.it (E.C.); gianluca.ottolina@scitec.cnr.it (G.O.); 3CREA Research Centre for Vegetable and Ornamental Crops (CREA OF), Corso degli Inglesi 508, 18038 Sanremo, Italy; marina.laura@crea.gov.it; 4Department of Agricultural and Environmental Sciences, University of Milan, Via Celoria 2, 20133 Milan, Italy; franco.faoro@unimi.it

**Keywords:** elicitation, *flax*, plant tissue cultures, justicidin B, methyl jasmonate, coronatine

## Abstract

Lignans are the main secondary metabolites synthetized by *Linum* species as plant defense compounds but they are also valuable for human health, in particular, for novel therapeutics. In this work, *Linum austriacum* in vitro cultures, cells (Cc), adventitious roots (ARc) and hairy roots (HRc) were developed for the production of justicidin B through elicitation with methyl jasmonate (MeJA) and coronatine (COR). The performances of the cultures were evaluated for their stability, total phenols content and antioxidant ability. NMR was used to identify justicidin B and isojusticidin B and HPLC to quantify the production, highlighting ARc and HRc as the highest productive tissues. MeJA and COR treatments induced the synthesis of justicidin B more than three times and the synthesis of other compounds. RNA-sequencing and a de novo assembly of *L. austriacum* ARc transcriptome was generated to identify the genes activated by MeJA. Furthermore, for the first time, the intracellular localization of justicidin B in ARc was investigated through microscopic analysis. Then, HRc was chosen for small-scale production in a bioreactor. Altogether, our results improve knowledge on justicidin B pathway and cellular localization in *L. austriacum* for future scale-up processes.

## 1. Introduction

Plants are source of a great number of organic compounds that do not directly participate in growth and development. The demand for these compounds, referred to as secondary metabolites, has increased in recent years due to their application in pharmaceutical and nutraceutical industries. Plant cell cultures were introduced in the late 1960s as a valid tool for the production of secondary metabolites [1]. Since then, the culture system has been extensively developed as shoots, calli, root and hairy root cultures that introduce new approaches such as elicitation and metabolic engineering to improve yield of bioactive compounds [2]. Plant cell cultures indeed provide an excellent system for the large-scale production, making in many cases possible to obtain and commercialize chemicals from rare plants. The major advantage of this method is the continuous, reliable source of bioactive molecules [3]. Among secondary metabolites, phenolic compounds form one of the major class. They are widely distributed in higher plants and they derive from the phenylpropanoid pathway [4]. These compounds play an important role in plant defense against pathogen attack, herbivores or environmental stresses, such as UV-irradiations, high light, wounding, nutrient deficiencies, temperature and herbicide treatment [5]. Moreover, they display several pharmacological activities such as anti-inflammatory, antioxidative, antimicrobial, antiviral and antineoplastic [6,7,8,9]. Among the various classes of phenylpropanoid compounds, lignans represent a large group with diverse biological activities [10]. Based on their chemical structure, they are divided into three main classes: aryltetralin-type, arylnaphtalene-type and dibenzylbutyrolactone-type lignans [11].

The genus *Linum* includes more than 200 species taxonomically divided into five or six sections [12,13]. Two sections of the genus *Linum* mainly accumulate arylnaphthalene-type lignans such us justicidin B [11,14]. Justicidin B exhibits several intriguing pharmacological activities such as antifungal (e.g., against *Aspergillus fumigatus*, *A. flavus* and *Candida albicans*), antiviral (e.g., against *Vesicular stomatitis virus*, VSV), antiparasitic (e.g., against *Trypanosoma brucei*), piscicidal, antiplatelet (e.g., playing a significant role in the prevention of atherosclerosis and thrombosis development), anti-inflammatory and cytotoxic (e.g., against chronic myeloid and chronic lymphoid leukemia line cells) [15]. Recently, it has been reported that justicidin B plays a positive effect on mesangial proliferative glomerulonephritis (MsPGN), one of the most common renal disease in China [16]. In *Linum*, production of justicidin B has been reported for in vitro cultures of *L. austriacum* [17], *L. narbonense* [18], *L. altaicum*, *L. lewisii* [19], *L. leonii* [20], *L. campanulatum* [21] and *L. glaucum* [22]. To date, the highest production of justicidin B has been obtained from hairy root cultures of *L. perenne* subsp. *himmelszelt* grown for 14 days (37 mg/g of justicidin B per dry weight of tissue, DW) [23].

Although the production of secondary metabolites is often low (less than 1% DW) [24], it could be improved by elicitation with substances that initiate or enhance the biosynthesis of a specific compound when introduced in small quantities in the living cell system [25]. Plant growth regulators, such as jasmonate derivatives, play many important roles in defense response by acting as signal molecules [26]. However, they could be effective elicitors when added exogenously in the culture medium [27]. Coronatine (COR), produced by pathovars of the plant bacterium *Pseudomonas syringae*, is a molecular mimic of the isoleucine-combined form of jasmonic acid [28]. The latter is a plant growth regulator and an elicitor that induces secondary metabolites in plants which has been reported to enhance the accumulation of taxane in *Taxus media* and *Corylus avellana* cell cultures [29,30].

In this work, we developed three in vitro tissue cultures from *L. austriacum*: calli (Cc), adventitious roots (ARc) and hairy roots (HRc) with the aim to select the most promising tissue along with the most promising elicitor treatment between methyl jasmonate (MeJA) and COR in term of justicidin B production. The total phenol contents and the antioxidant activity of cell and root extracts were evaluated, and justicidin B was identified by NMR spectroscopy and quantified by HPLC analysis. The expression profile of genes activated by MeJA elicitation in ARc was investigated by RNA-sequencing and a de novo assembly of *L. austriacum* transcriptome was generated. Furthermore, we investigated, for the first time, the intracellular localization of justicidin B in ARc of *L. austriacum* through microscopic analysis.

Finally, a first explorative experiment using a small-scale bioreactor has been performed with HRc of *L. austriacum*. The results obtained improve the knowledge on justicidin B pathway for future scale-up processes.

## 2. Results

### 2.1. Tissue Cultures, Elicitors Treatment and Phenolic Content

Three different lines of calli, three of adventitious roots and seven of hairy roots were obtained from *L. austriacum*. Examples of each tissue obtained are showed in Figure 1a. The growth performance of each line of calli was assessed to select the most promising. Upon inoculum in liquid medium, cell cultures (Cc) were grown for a maximum of 14 days and the growth was followed by measuring the cell volume after sedimentation (CVS). The fresh-weight (FW) and the dry-weight (DW) were also monitored. The growth curve of the most promising cell culture (Cc line 1) is reported in Figure 1b. After a lag phase that lasted until day 6, the exponential phase occurred from day 6 to day 11, then a stationary phase with a decrease of the growth was observed together with the browning of the culture and the formation of clumps, both of which are symptoms of ageing [31]. During the lag and the exponential phases, the cell viability was greater than 80%.

Since secondary metabolites are mostly synthetized during exponential phase [32,33], the elicitor treatments were applied to Cc just entered in this period. Before starting the elicitor treatments, the percentage of viability was evaluated, and 80–90% rate of success was considered acceptable to proceed with the experiment. It was observed that both methyl jasmonate (MeJA) and coronatine (COR) caused an inhibitory effect of 20–25% on the cell growth after four days of treatment (Figure 1b).

PCR analysis on the different hairy root culture lines confirmed that the *rolC* gene of *Agrobacterium rhizogenes* had been transferred into the DNA of all samples tested. Indeed, lack of amplification of the virC1 fragment confirmed the absence of any contaminating *A. rhizogenes* sequence in all the hairy root culture lines analyzed.

Upon inoculum in liquid medium, the growth kinetics of both adventitious root (ARc) and hairy root (HRc) culture lines was estimated up to one month to evaluate the most promising line (Appendix A). Then, the ARc line 2 and the HRc line 7 were selected for all the experiments hereinafter and their growth kinetics are reported in Figure 1c,d, respectively. In detail, the growth curve of ARc showed a lag period lasting one week and an exponential phase occurring during the second and the third weeks, then the cultures entered a stationary phase (Figure 1c). A similar behavior was observed for HRc Figure 1d. The roots morphology and color remained the same throughout the experiment (Appendix A). Based on this kinetic, the elicitor treatments started 8/10 days after the inoculum of the ARc and HRc in liquid medium, corresponding to the exponential phase, and the roots were allowed to grow for another four days. The growth rate of ARc and HRc evaluated in presence of the elicitor showed that both MeJA and COR caused an inhibitory effect of about 21–23% on the growth rate (Figure 1c,d).

### 2.2. Total Phenolic Content and DPPH Radical Scavenging Activity

Total phenols content and 2,2-diphenyl-1-picrylhydrazyl (DPPH) scavenging activity were determined on control, MeJA- and COR-treated samples to better characterize the three different tissue culture systems obtained. As shown in Figure 2a, COR elicitation induced the highest accumulation of total phenols in each tissue examined. Both COR-treated Cc and ARc almost doubled the total phenols content over the control, whereas in COR-treated HRc the phenols content was 1.5 times higher than the control. However, the HRc control showed a double phenol content compared to the control of Cc and ARc. Moreover, the COR-elicited HRc exhibited the highest phenol content reaching 9.84 ± 0.48 µg gallic acid equivalent per mg dry weight (µg GAE/mg DW). Notably, MeJA elicitation was more effective on ARc and HRc than on Cc where the difference caused by the treatment was less significant.

The effect of elicitation was also evaluated on the antioxidant capacity of the extracts of the three different tissue cultures (Figure 2b). In all the samples analyzed, the free radical scavenging activity increased after elicitation, particularly with COR treatment. The maximum antioxidant activity (expressed as percentage of DPPH inhibition per mg of DW) was observed in COR-treated HRc (27.80 ± 3.25%). Moreover, HRc elicited with MeJA showed a statistical significant increase (1.4-fold) of free radical scavenging with respect to the control.

### 2.3. Isolation, Identification and Quantification of Justicidin B

To isolate justicidin B, a thin layer chromatography (TLC) was performed on control and treated (MeJA and COR) Cc, ARc and HRc extracts exploiting its fluorescence at 366 nm [34]. The analysis showed the presence of a main fluorescent band in the control samples and two additional bands in all the elicited samples (Figure 3a). The most intense fluorescent band was collected and analyzed by ^1^H-NMR to investigate its identity. The chloroform solution was analyzed by using both homo- and heteronuclear multidimensional experiments. The ^1^H-NMR revealed the presence of a unique compound (Figure 3b) and ^1^H-^13^C HSQC spectra of the extract (Figure 3c) confirmed the full proton and carbon assignment for justicidin B (Appendix A). This assignment resulted in agreement with previously reported data [35]. In detail, the characteristic singlets at 7.72, 7.20 and 7.13 ppm are indicative of a double substituted arylnaphthalene ring, leaving three isolated protons. Interestingly, the characteristic non-isochronous germinal protons in position 10 give rise to a non-unique chemical shift value, most likely due to their relative position with respect to the aromatic ring (Figure 3c).

High performance liquid chromatography (HPLC) was used to quantify the amount of justicidin B in control and in Cc, ARc and HRc elicited samples. As shown in Figure 4, COR-elicited ARc was the most productive tissue of justicidin B (15.74 ± 1.18 mg/g DW). On the other hand, among tissues elicited with MeJA, the most productive was HRc (14.71 ± 0.90 mg/g DW). Cc was the tissue with the lowest production of justicidin B.

Indeed, HPLC analysis revealed the presence of other two molecules besides justicidin B, named m2 and m3 (Figure 5a). Therefore, the methanolic ARc elicited extract was subjected to a further purification to fully remove the justicidin B, and the residual was investigated by ^1^H-NMR (Figure 5b). Accordingly, the analysis revealed the presence of different molecules and through ^1^H-^13^C HSQC experiment, the most abundant one (m3) was identified as isojusticidin B (Figure 5b,c). This arylnaphthalene type lignan was identified through the characteristic “naphthalene-type” doublets due to vicinal aromatic protons in positions 5 and 6 in the aromatic region of the ^1^H-NMR spectrum. Additionally, the chemical shifts of two methoxyl groups at 3.97 and 3.35 ppm confirmed the “iso” position of the methoxyl groups, due to the upfield shifted effect of the aromatic ring on the methyl 11. Further observed in this molecule is the characteristic non-isochronocity of the geminal protons in position 10. The full proton and carbon assignment for isousticidin B is reported in Appendix A. The remaining signals present in both the aromatic and the aliphatic region of the ^1^H-NMR spectrum strongly suggest the presence of other molecules, not yet identified. In this case, the NMR investigation provided solid ground for the establishment of the iso conformation for justicidin B, something that was not possible with other analytical methodologies.

In Figure 6 are shown the abundance of each molecule in the three tissue cultures. In agreement with Figure 4, justicidin B was mainly produced by MeJA-elicited HRc and COR-elicited ARc. Isojusticidin B and m2 were mainly produced by MeJA- and COR-elicited HRc and COR-elicited ARc. Cc produced all the three molecules but at very low level, in particular isojusticidin B and m2.

### 2.4. RNA-Seq Analysis

To identify genes involved in the justicidin B pathway that are possibly induced by MeJA, RNA-seq transcriptional profiling was performed on ARc collected at four different time points after elicitation: 2, 5, 24 and 96 h. Because of the lack of a *L. austriacum* reference genome, a de novo transcriptome assembly was generated and used for expression analysis. Overall, an important number of transcripts resulted differentially expressed (DETs) between the control and the elicited samples, and down-regulated genes exceeded in number those up-regulated in all time points (Figure 7). The greatest number of DETs was observed after 2 h from elicitation, although their number remained high through the different time points, suggesting that major alterations on gene expression were present through the course of the whole experiment (Figure 7a). On the whole, 15% (5380 at 2 h), 5.30% (1899 at 5 h), 9.10% (3262 at 24 h) and 10% (3605 at 96 h) of total DETs were time-point-specific, whereas 13% (4804) of total DETs were common to all time points (Figure 7b).

The pathway leading to the justicidin B production can be divided in three main steps, each composed by different reactions (Figure 8a and Appendix A). Using the sequence information available at the Plantcyc database (https://plantcyc.org/content/flaxcyc-3.0, accessed on 20 January 2021) for *L. usitatissimum*, for other plant species (such as *Arabidopsis thaliana*, *Coffea arabica*, *Oryza sativa*, *Forsythia intermedia*, *Podophyllum peltatum*, *Glycine max* and *L. perenne*) and from previous works [36,37,38,39,40,41,42,43,44,45,46], representative transcript sequences of each reaction were retrieved when present. Then, these sequences were used to identify, by BLASTP, putative orthologue transcripts in *L. austriacum* transcriptome (Appendix A).

Collectively, 63 transcripts encoded by 51 *loci* encompassing 14 reactions of the pathway were identified as differentially expressed in at least one comparison with the control sample based on their significance (false discovery rate: FDR < 0.05) and expression difference level (log_2_FC ≥ 1; Figure 8b). Although transcript targets were identified for all reactions involved in step 1 and 2, for step 3 transcript targets were obtained only for five reactions. Specifically, the first reaction of this step was not considered despite the large number of hits present in our transcriptome, as this step is catalyzed by either a laccase or a peroxidase enzyme. Moreover, to date, no information is available for the last reactions of the pathway leading to justicidin B synthesis starting from matairesinol (Figure 8a). Notably, the reactions R3 and R4, R5 and R6 of step 3 are supposed to be catalyzed by the same set of enzymes.

In contrast with the general transcriptional trend of the samples analyzed, DETs associated with the pathway were preferentially up-regulated after 2 and 4 h. Namely, at two and four hours the 48% and 36% genes were up-regulated vs. 16% and 31% down-regulated, respectively.

After 24 h of treatment, the highest number of DETs (81%) was observed, and up- and down-regulated DETs showed similar percentages (40%). Onwards, after 96 h, the total number of DETs was the lowest (49%), with down-regulated transcripts slightly exceeding the up–regulated ones, i.e., 27% vs. 22%.

The first step of the pathway includes two reactions that lead to the production of 4-coumarate starting from L-phenylalanine and are catalyzed by the phenylalanine ammonia-lyase and trans-cinnamate 4-monooxygenase (Figure 8a). After two hours of elicitation all transcripts identified (5) in this step were up-regulated (Figure 8b), and this trend was generally persistent through the four time points.

The second step of the pathway includes seven reactions that lead to the formation of coniferyl alcohol using the 4-coumarate as substrate (Figure 8a). In this step, both up- and down-regulated transcripts (36) were observed in six out of seven reactions. In particular, in R1, R3, R6 and R7, which are catalyzed by the 4-coumarate-CoA ligase, p-coumaroyl ester 3’-hydroxylase, cinnamoyl-CoA reductase and the coniferyl alcohol dehydrogenase, respectively (Figure 8a), only 5 out of 17 transcripts were up-regulated in at least two time points, and in this step only one was stably up-regulated in all time points (R5, Figure 8b).

The final step of the pathway includes reactions that are specific to plant-producing lignans. A total of 22 transcripts was identified for reactions that range from R2 to R6 (Figure 8a), which lead to matairesinol starting from pinoresinol: as previously mentioned, R3/R4 and R5/R6 are supposed to be catalyzed by the same type of enzymes. Compared with other reactions of the pathway, R2 includes the largest number of transcripts identified (15) within the *L. austriacum* transcriptome. These transcripts, which putatively encode for a dirigent protein, were largely down-regulated in all time points, and only 2 out of 15 were constantly up-regulated (Figure 8b). A nearly constant up-regulation was observed for the three transcripts associated to R3/R4, which are controlled by the pinoresinol reductase, whereas up-regulation was observed for two out of four transcripts associated to R5/R6, which encode for a secoisolariciresinol dehydrogenase (Figure 8b). Assuming a positive correlation between justicidin B accumulation and gene expression level, a tentative list of transcripts likely involved in justicidin B pathway was determined (Appendix A).

### 2.5. Microscopic Analysis of Justicidin B in Adventitious Roots

ARc of *L. austriacum* examined by conventional fluorescence microscopy with UV light showed a bright blue autofluorescence usually starting from the end of the elongation zone and particularly evident in the maturation area (Figure 9). At low magnification (Figure 9a) it appeared that fluorescence was significantly higher in MeJA-elicited roots with respect to the non-elicited ones, while no autofluorescence was present in *L. mucronatum* roots of the same age used as negative control. *L. mucronatum* was chosen as negative control since it belongs to a *Linum* section producing aryltetralin-type lignans [11,14]. At higher magnification, fluorescence appeared concentrated in single spots inside cells (Figure 9b) and was visible only with the excitation filter of 330–385 nm but not with higher wavelengths, confirming that it was due to justidicin B deposition inside cells. To improve the resolution of these spots and a more precise localization of justidicin B, a confocal laser scanning microscope (CLSM) examination was carried out, setting the laser at the maximum excitation of this compound (366 nm). As illustrated in Figure 9c, CLSM revealed that the spots visible by conventional fluorescence were in turn formed by much smaller spots mainly located on the periphery of the cortical parenchymatic cells of the root, thus likely in the cytoplasm, but not on the wall or in the vacuole, suggesting that justidicin B is not secreted, at least in this growth condition.

### 2.6. Justicidin B Production in Bioreactor

HRc line 7 was chosen for the scale-up production of justicidin B in a bioreactor (Figure 10) due to its greater adaptability and better growth kinetics than the other tissue cultures obtained. In particular, HRc line 7 was used being the most performing in term of growth Appendix A and justicidin B production. In the bioreactor, HRc line 7 gave rise to a 5-fold increase biomass production after 21 days of growth with a production of 21.30 mg of justicidin B, giving then 21 mg/g DW. During a longer growth period (35 days) in the bioreactor, the HRc line 7 reached a 14-fold increase in biomass with a production of 35.60 mg justicidin B, namely 9 mg/g DW.

## 3. Discussion

Justicidin B exhibits a wide array of biological properties and is considered a potential lead compound for novel therapeutics [15]. Although significant progresses in the chemical synthesis of arylnaphthalene lactones have been made, there are still significant limitations due to its economic unsustainability. However, in vitro cultures represent an interesting alternative for scale-up purposes. In this study, cell (Cc), adventitious roots (ARc) and hairy roots (HRc) in vitro cultures were developed from *L. austriacum* to select the most promising tissue in term of justicidin B and lignans production. In agreement with data obtained from *L. austriacum* and other plant species [17,47], we found that the ARc and HRc were the best producer tissues based on total phenol content, antioxidant ability and amount of justicidin B. However, slight differences in justicidin B production were observed between our tissues and those analyzed by Mohagheghzadeh et al. [22] that might be due to the different seed origin (Armenia versus Iran) and to the different growth rate of the two systems.

### 3.1. Elicitation, Time Course of Growth, Phenols Accumulation and Antioxidant Activity

Elicitation with different molecules, e.g., methyl jasmonate (MeJA) or coronatine (COR), has been described as a strategy to enhance secondary metabolites production [29,48]. When MeJA is exogenously applied to plant cell cultures, it positively stimulates the secondary metabolites’ pathway via signal transduction, leading to the formation of specific compounds, such as polyphenols, terpenoids, flavonoids and alkaloids [49]. COR, a toxin produced by the pathogen Pseudomonas syringae, was first described as elicitor by Weiler et al. [50] who observed that the treatment of the in vitro culture with COR is able to induce secondary metabolite biosynthesis even though at lower concentration than MeJA. In our experiments, the elicitor treatments were performed during the exponential growth culture phase as it was reported that secondary metabolites are mostly synthetized after the culture growth enters this phase [33,51,52]. Similarly to what was described for several plant tissue cultures [29], in our experiments the growth of both cell and root biomass significantly decreased (21%) after the elicitation. In this regard, it has been proposed that both MeJA and COR presumably cause an arrest in the cell cycle progression at the G2-phase [53,54]. Indeed, elicitor treatments might have resulted in the production of reactive oxygen species (ROS), which negatively affect cell and root biomass; then, to mitigate the ROS effects, the plant increased the production of secondary metabolites [55,56]. Accordingly, the elicitor treatments increased the total phenol content in each tissue culture developed. In particular, HRc showed the highest phenols’ content, whereas Cc and ARc exhibited a similar phenolics level. Moreover, the elicitor treatments induced a higher total phenolic accumulation in each tissue analyzed, similarly with reports by Ramirez-Estrada et al. [57]. Considering the antioxidant activity described for phenolic compounds [58], the DPPH radical scavenging activity of the study tissues was also evaluated. We found that antioxidant capacity of the extracts from the three tissues correlated to their phenolic content. In particular, HRc showed the highest antioxidant power among the examined tissues, and COR-treated HRc were the most effective. These results are in agreement with previous reports on flax seeds, hairy roots and extracts from other medical plant species [59,60,61].

### 3.2. Identification, Isolation and Quantification of Justicidin B

One of the main lignans described for *L. austriacum* is justicidin B [15]; thus, we focused on the identification, isolation and quantification of this compound in the three tissue cultures developed. This molecule was detected in each tissue culture, but ARc and HRc produced four times more justicidin B (3.94 and 4.89 mg/g DW, respectively) than the conventional calli cultures. Moreover, ARc achieved the maximum content of justicidin B when elicited with COR (15.74 mg/g DW) whereas the HRc reached the highest content when elicited with MeJA (14.71 mg/g DW). Indeed, it has been observed in *Salvia sclarea* hairy roots that MeJA was more effective in triggering aethiopinone accumulation than COR [52]. As far as the content of justicidin B is concerned, we obtained lower amounts than those reported by Mohagheghzadeh et al. [17] for *L. austriacum* hairy roots (16.90 mg/g DW). As above mentioned, this discrepancy could be due to the different origin of the genotypes used for the experiments and to the different growth rate. Our data also showed that the amount of justicidin B produced by COR-elicited ARc and MeJA-elicited HRc were comparable; nevertheless, HRc exhibited better growth kinetics in terms of biomass accumulation, fast growth rate, genetic and biochemical stability, relatively simple maintenance in phytormone-free media and increased lateral branching. Indeed, HRc provide a higher yield of secondary metabolites (particularly when elicited), in line with previous reports [62]. In addition, Hemmati et al. [23] demonstrated that in vitro cultures obtained from *L. perenne Himmelszelt* (same section with *L. austriacum*), produced other molecules besides justicidin B. Indeed, HPLC analysis of our extracts showed the presence of other two molecules (named m2 and m3). Among these, isojusticidin B (m3) was identified by NMR, whereas the identification of m2 is still in progress. In essence, justicidin B, isojusticidin B and a not identified compound (m2) were induced by elicitor treatments. In particular, isojusticidin was produced mainly by the COR-treated HRc, whereas m2 level was similar in the COR-treated HRc and Arc.

### 3.3. Bioreactor Culture

Based on these results, the HRc cultures were chosen to test preliminary the feasibility of a production up-scale (one liter bioreactor). Although few works have described the production of justicidin B from HRc of *Linum* spp. [18,63], the use of *L. austriacum* roots for the production of justicidin B in a bioreactor system has not previously reported to the best of our knowledge. Among different HRc clones available, HRc line 7 was chosen because of its greater adaptability and better growth kinetics. Our results showed that the amount of justicidin B produced in the bioreactor by the HRc line 7 grown for 21 days was significantly higher than the amount produced by HRc in the flask for the same time (21.30 ± 1.29 and 4.89 ± 0.89 mg/g DW, respectively). This preliminary result gave an estimated productivity of 1.56 mg/L d obtained without the addition of the elicitor. The production yield of justicidin B obtained from HRc line 7 in our bioreactor system was higher than that reported for *L. narbonense* and *L. leonii*, namely, 14.20 mg/g DW and 15.50 mg/g DW, respectively [18,63]. We found that a longer growth period (35 days) did not give rise to a proportional enhancement of justicidin B production even when the biomass was further increased. As previously reported, the culture entered into the stationary phase after three weeks and, as a result, a metabolic change may have taken place [64].

### 3.4. Transcriptome Analysis of Justicidin B Pathway

Unlike *L. usitatissimum* [65] and references therein, scarce information is available at transcriptional level for *L. austriacum*, thus limiting the identification and characterization of transcripts involved in different biological processes of interest. To identify gene encoding enzymes associated to the justicidin B pathway, we performed an RNA-seq analysis on ARc samples elicited with MeJA. The choice of this tissue-elicitor combination was determined by two main reasons. The first is that ARc is not a transformed tissue, contrary to HRc that may produce genetic variability, such as T-DNA copy number insertion, having a direct impact on gene expression data. Secondly, the choice of MeJA as elicitor was due to the better characterized regulatory mechanism and biological activities triggered by this elicitor with respect to COR, especially on ARc tissue culture. In particular, a study comparing COR- and MeJA-regulated transcriptomes in tomato revealed that COR regulated 35% of the MeJA-induced genes [66].

Given the lack of reference genome, a de novo assembly of *L. austriacum* transcriptome was generated. This represents the first genome-wide data produced for this species to the best of our knowledge. Overall, a total of 60,041 transcript isoforms associated to nearly 32,252 genes were identified. At gene level, the total number of genes identified by our analysis is lower than those identified in the reference species *L. usitatissimum*, based on in silico prediction of genomic assembly data: i.e., 43,484 genes [67]. However, this discrepancy could be explained since our analysis focused on transcripts generated in a single tissue subjected to specific growth conditions, which may not be representative of the whole transcriptome of this species. Indeed, in agreement with our results, transcriptome analysis of different developmental stages of a drought tolerant cultivar of *L. usitatissimum* identified a total of 61,563 transcripts clustered into 39,330 genes [68]. Furthermore, evidences of variability in total transcripts, likely due to sample type, are provided by transcriptome analysis of *L. usitatissimum* leaves exposed to different nutritional conditions, in which between 33,698 and 34,924 unique transcripts were identified [69]. Thus, although our analysis might still represent a preliminary view of the transcriptomic complexity of *L. austriacum*, it provides one of the first genome-wide data set of this species useful for further studies.

At first glance, ARc induced with MeJA displayed a remarkable level of differentially expressed transcripts (DETs) that reached their maximum after two hours of elicitation. However, high level of DETs were also observed after 96 h, suggesting that MeJA had an important impact on gene expression regulation of ARc, and that this effect is persistent across the whole experiment. These results agree with other studies analyzing the effect of MeJA in several plant species [53,70,71] where evidence of massive gene expression reprograming following elicitation has been reported.

In this study, we specifically focused on candidate transcripts involved in the justicidin B pathway, and a more detailed characterization of transcriptome data will be part of a future study. Starting from the L-phenylalanine, 63 transcripts encoded by 51 genes involved in the 14 downstream reactions leading to matairesinol were identified as DETs. As expected, MeJA preferentially up-regulated genes involved in justicidin B pathway, and this occurred mainly within the first 24 h. Nevertheless, down-regulation of several transcripts of different reactions occurred in both step 2 and step 3. For instance, in the reaction R7 of step 2, in which coniferyl alcohol dehydrogenase (CAD) catalyzed the conversion of coniferaldehyde to coniferyl alcohol, none of the transcripts identified were constantly up-regulated throughout the whole experiment. This result was not unexpected considering that the transcript belongs to a large gene family just like other genes involved in justicidin B pathway, and that the expression of these genes could be both up- or down-regulated depending on species, tissue examined or the elicitor used [36]. Similar observations were reported from other genes of the pathway investigated across both tissues and stress condition [37,38]. For example, the six *L. usitatissimum CAD* genes found up-regulated in flax Cc treated with MeJA [36], showed identities with only two up-regulated transcripts in *L. austriacum* (Appendix A). Likewise, Corbin et al. reported that in *L. usitatissimum* HRc elicited with MeJA, several DIR transcripts were positively regulated [38]. Expression analysis in *L. austriacum* ARc revealed that only two dirigent protein transcripts were up-regulated, being *DIR1, DIR2, DIR3* and *DIR37*, most likely the orthologues [38] (Appendix A). Given that MeJA has a promoting effect on the Cc, HRc and ARc in terms of justicidin B induction, the transcriptional differences observed between different studies may help to discriminate between allelic variants directly involved in the justicidin B pathway from others which are not.

### 3.5. Justicidin B Cellular Localization

Since a technology for the production of plant secondary metabolites is based on the extraction via exudates or release [72], we examined the intracellular localization of justicidin B considering that, as far as our literature survey could ascertain, no prior information was available. Then, microscopic analysis of *L. austriacum* ARc was performed exploiting the fluorescence of this molecule. The confocal analysis highlighted that justicidin B was mainly located on the periphery of the cortical parenchymatic cells of the root, thus likely in the cytoplasm, but not on the wall or in the vacuole, suggesting that justidicin B is not secreted, at least in this growth condition. Indeed, in the growth medium of the examined tissues, justicidin B was lower than 1% of that extracted from ARc and HRc (data not shown). Apart from indicating that all justicidin B can be recovered mainly from roots extracts, the findings of this study will be helpful in directing future studies aimed at the biotechnological production of this metabolite.

## 4. Conclusions

Three different in vitro cultures were developed from *L. austriacum* and the effectiveness of two elicitors, methyljasmonate (MeJA) and coronatine (COR), on the production of justicidin B were evaluated. We found that the adventitious roots (Arc) and hairy roots (HRc) were the best producer tissues based on total phenol content, antioxidant ability and amount of justicidin B. To better elucidate the justicidin B pathway and the differential expressed genes after elicitation, Arc were chosen to produce a de novo transcriptome assembly for *L. austriacum* species. The results obtained together with those regarding the cellular localization of justicidin B will enrich the knowledge for future scale-up process. Indeed, in this study a small-scale bioreactor was tested with *L. austriacum* HRc, which have a fast growth rate, are genetically stable and are simple to maintain in phytohormone-free media.

## 5. Materials and Methods

### 5.1. Plant Material and Cultures

*L. austriacum* seeds were obtained from USDA (U.S. Department of Agriculture). The seeds were surface sterilized in 70% (*v*/*v*) ethanol for 1 min, then in sodium hypochlorite diluted 1:5 in water (*v*/*v*), washed 5 times in sterilized water, and the germination occurred in phytatray (Merck, Darmstadt, Germany) on Murashige and Skoog basal medium (MS, Duchefa, Haarlem, The Nederlands), [73] at 22 °C in dark conditions. After one week the seedlings were placed at 25 °C under 16 h light and 8 h darkness for one month.

To induce calli, abaxial surface leaf explants were inoculated on MS-medium supplemented with 1 mg/L (*w*/*v*) α-naphthalene acetic acid (NAA), 0.5 mg/L (*w*/*v*) kinetin (KIN) and 0.5 mg/L 2,4-dichlorophenoxyacetic acid (2,4-D). All media mentioned were supplemented with 3% (*w*/*v*) sucrose and solidified with agar 0.8% (*w*/*v*) at pH 5.8 before autoclaving. After callus production and culture establishment, calli were routinely subcultured every three weeks and maintained under permanent dark conditions at 25 °C. To initiate cell suspension cultures (Cc), 1.5 g fresh weight (FW) of calli were transferred to a 250 mL Erlenmeyer flask containing 50 mL of MS liquid medium. Cc were grown at 25 °C on an orbital shaker at 110 rpm under permanent dark conditions.

Adventitious roots were generated from leaves collected from one month-old plantlets of *L. austriacum*. The abaxial surface leaf explants were transferred to MS medium supplemented with 0.4 mg/L NAA and 1 mg/L 2,4-D. Early root primordia emerged after 15–20 days of culture. The response of the explants to the formation of adventitious roots was found to be 90%. After about four weeks, the length of the root segments was between 0.8 and 1.3 cm. At this stage, roots were individually transferred to MS medium supplemented with 0.5 mg/L indole-3-butyric acid (IBA) and 0.1 mg/L indole-3-acetic acid (IAA) (shortened from now on as MS-II medium) in order to increase their biomass and length, and they were subcultured every month. To initiate adventitious root suspension culture (ARc), 0.8 g FW of adventitious roots were transferred to 250 mL Erlenmeyer flasks containing 50 mL of liquid MS-II medium. ARc were grown at 25 °C on an orbital shaker at 110 rpm under permanent dark conditions.

To induce hairy roots formation, leaf and stem explants from in vitro seedlings of *L. austriacum* were incubated with *Agrobacterium rhizogenes* strain ATCC 15,834 [74]. The *A. rhizogenes* was grown overnight at 28 °C in yeast mannitol broth (YMB: 0.5 g/L K_2_HPO_4_, 0.2 g/L MgSO_4_·7H_2_O, 0.1 g/L NaCl, 10 g/L mannitol and 0.4 g/L yeast extract) supplemented with 200 μM acetosyringone (AS). The cultures were then pelleted by centrifugation and resuspended in MS medium supplemented with 200 μM AS and adjusting OD_600_ to 0.6. The explants were incubated in the *Agrobacterium* solution for 20 min at room temperature and then blotted on sterile filter paper. The explants were co-cultivated with *Agrobacterium* on solid co-cultivation media (MS medium with 3% (*w*/*v*) sucrose and 0.8% (*w*/*v*) agar, pH 5.7) for 3 days at 24 °C in the dark and then were transferred on the same media supplemented with 100 μg/mL cefotaxime. After about two weeks, calli were formed from all explants and then from two weeks-old calli several hairy roots emerged. Individual putative hairy roots of approximately 3 cm long were excised and immediately transferred on MS medium supplemented with 50 μg/mL cefotaxime. All the hairy roots were maintained separately as independent lines. The roots were subcultured every 15 days in the presence of 50 μg/mL cefotaxime until the complete elimination of bacteria. To initiate hairy root suspension culture (HRc), 0.8 g FW of hairy roots were transferred to 250 mL Erlenmeyer flasks containing 50 mL of MS medium. HRc were grown at 25 °C on a gyratory shaker at 110 rpm under permanent dark conditions. All the media and components for in vitro cultures were purchased by Duchefa-Biochemie, Haarlem, The Netherlands.

Genomic DNA from 10 independent lines of hairy roots were extracted using DNeasy Plant Mini Kit (Qiagen, Hilden, Germany) according to the manufacturer’s instructions. The genomic DNA extracted was used as template to amplify a fragment of rolC gene of *A. rhizogenes*. The primer sequences are as follows: Forward: 5′-CGACCTGTGTTCTCTCTTTTTCAAGC-3′ and Reverse: 5′-GCACTCGCCATGCCTCACCAACTCACC-3′.

The lines that were positive in PCR analysis of the rolC gene were checked for the absence of *A. rhizogenes* DNA by a PCR targeting the 326 bp fragment of virC1 (bacterial chromosome), following [75].

### 5.2. Growth Measurement and Kinetics of Cc, ARc and HRc

To determinate the Cc growth in each flask, cell volume sedimentation (CVS) was measured, then 0.5 g of fresh calli was cultured in 250 mL flask with 50 mL of MS medium for two weeks and sampled at fixed time. This method is rapid and simple and allows the routine estimation of cell biomass, without the destruction of cells. Moreover, CVS is highly correlated with the FW of cells [76]. The cell viability was assessed by Evans blue (Sigma-Aldrich, St. Louis, MO, USA) vital exclusion dye assay [77]. In detail, 20 µL of 10% (*w*/*v*) Evans blue was added to 300 µL of Cc, vigorously shook and place 15 min at room temperature. The excess of stain was removed from the medium with distilled water, whereas the stain bound to dead cells was solubilized in 50% (*v*/*v*) aqueous methanol containing 1% (*w*/*v*) SDS and quantified spectrophotometrically by measuring the OD_600_. Heat treated cells (boiling 10 min at 100 °C) were used as control of 100% cell death.

The growth of both ARc and HRc was measured as FW. Roots were harvested, washed three times with sterilized distilled water, dried on filter paper to remove all external moisture and weighed. FW was expressed in grams/flask (g/flask). For the determination of root viability, a solution of 10 mM KH_2_PO_4_, 3% (*w*/*v*) sucrose, 0.25% (*w*/*v*) triphenyl tetrazolium chloride (TTC) was used as a visual indicator of root viability [78]. ARc and HRc growth was monitored up to one month and the FW was determined every week. Each time point was replicated three times and the starting material was set at 0.8 g of fresh tissue.

### 5.3. Elicitor Treatments

Suspension cultures of Cc, ARc and HRc were elicited with 100 μM methyljasmonate (MeJA, Merck, Darmstadt, Germany) and 10 μM coronatine (COR, Merck, Darmstadt, Germany) [29,52,79]. MeJA was dissolved in absolute ethanol and sterilized by filtration (0.22 µm), COR was dissolved in distilled water. The respective controls were performed supplementing the medium with ethanol at the same final concentration or water. Elicitor treatments of Cc started when the cultures entered into exponential phase (on day 7/8 from the inoculum in liquid medium), whereas the elicitor treatments of ARc and HRc started on day 10 after the initiation of the liquid culture. The cultures were further incubated for four days. The harvested cells or roots were grounded in liquid nitrogen, lyophilized and stored at −80 °C until analysis.

### 5.4. Lignans Extraction

Lignans extraction was performed from powdered lyophilized tissues. The samples were dissolved in 80% (*v*/*v*) methanol and homogenized with Ika Ultra Turrax T18 for 2 min, vortexed, sonicated for 10 min and extracted overnight on a giratory shaker at room temperature in the dark. Extracts were then centrifuged for 20 min at 13,000× *g* and the clear supernatants were used for total phenolics determination, 1,1-diphenyl-2-picrylhydrazyl (DPPH, Merck, Darmstadt, Germany) radical scavenging assay and chromatographic analysis.

### 5.5. Total Phenolic Content and DPPH Radical Scavenging Activity

The amount of total soluble phenolics was evaluated by Folin-Ciocalteu assay using gallic acid (GA) as standard, and total phenols were expressed as μg of GA equivalent per mg of DW (µg GAE/mg DW) [80]. In detail, 50 µL of the 80% (*v*/*v*) methanol extract solution were mixed with 750 µL of water and 50 µL of 10-fold diluted Folin-Ciocalteu reagent (Sigma-Aldrich) and allowed to stand 10 min at room temperature before 150 µL of 20% (*w*/*v*) sodium carbonate solution was added to each sample, and placed in the dark at room temperature for two h. The absorbance was spectrophotometrically measured at 765 nm. Folin-Ciocalteu and gallic acid were purchased by Merck, Darmstadt, Germany. The DPPH radicals scavenging assay was determined following the described method [81]. Briefly, a working solution of 0.208 mM DPPH in methanol was made daily and mixed with 30 µL of the extracts. The solution was incubated for 30 min at room temperature in dark conditions. The absorbance of the mixture was then spectrophotometrically measured at 515 nm against a blank of 80% (*v*/*v*) methanol. The ability to scavenge DPPH was calculated as indicated in Equation (1):(%) DPPH radical scavenging activity = [(A_ct_ − A_sa_)/A_ct_] × 100(1)
where A_ct_ is the absorbance of DPPH radical plus methanol and A_sa_ is that of DPPH radical plus the sample extract. Radical scavenging activity is shown as the percentage of DPPH inhibition for mg of DW.

### 5.6. Chromatographic Analysis

The methanolic extracts were dried under nitrogen and then dissolved in chloroform. The extracts were then evaluated by thin layer chromatography (TLC) and high performance liquid chromatography (HPLC).

#### 5.6.1. TLC experimental conditions

Qualitative analyses were carried out using TLC chromatographic separation following the method described previously [34]. The chloroform extracts were subjected to TLC separation on pre-coated silica gel glass plates 60 (Merck, Darmstadt, Germany) 10 × 20 cm, 2 mm thickness. The mobile phase consisted of methanol–chloroform (1:99, *v*/*v*). The chamber was pre-saturated for 15 min at room temperature. Justicidin B was revealed through its fluorescence at 366 nm [34]. The corresponding band was pooled and resuspended in chloroform.

#### 5.6.2. Analytical HPLC

HPLC analyses were carried out using a Jasco instruments equipped with photodiode array detector. Separation was performed using a Synergi polar RP 80 Å (250 mm × 4.60 mm, 4 μm, Phenomenex, Torrance, CA, USA) and a gradient system with water 0.1% (*v*/*v*) formic acid (A) and acetonitrile 0.1% (*v*/*v*) formic acid (B) as eluent as follows: 0–70% B for the first 26 min, from 70% to 0% B in 1 min and 0% B up to 30 min. The flow rate was 1.0 mL/min. The wavelength used for the integration of the signals was 254 nm, justicidin B retention time (t_R_) 26.3 min.

The justicidin B quantification was performed with standard calibration curve obtained using seven standard dilutions ranging from 9.13 to 86.00 μg/mL. Each standard solution was injected in duplicate. The linear regression equation was carried out by plotting the peak areas against the injected amounts of standard compounds, giving a R^2^ of 0.9990. The limits of detection (LOD) and the limits of quantification (LOQ) were determined as: LOD = 1.94 µg/mL and LOQ = 6.45 µg/mL.

#### 5.6.3. Semi-preparative HPLC

The purification of justicidin B by HPLC occurs in condition slightly different from the analytical runs. The sample was pooled in several batches and subjected to Synergi polar RP 80 Å (250 mm × 10 mm, 4 μm, Phenomenex, Torrance, CA, USA) by means of water 0.5% (*v*/*v*) formic acid and acetonitrile 0.5% (*v*/*v*) formic acid as eluents. The justicidin B obtained reached a purity up to 98.80% and was used as standard for HPLC quantification in analytical method.

#### 5.7. ^1^H-NMR Analysis

NMR spectra have been recorded at 14.09 T with a Bruker DRX spectrometer operating at 600 MHz equipped with a 5 mm reverse probe operating at room temperature. ^1^H-NMR spectra have been acquired quantitatively by using 128 scans with 8000 Hz of sweep width over 32 K points, at 298 K. ^1^H-^13^C heteronuclear bidimensional experiments, (SQC, HMBC) were recorded with 2 K and 256 data point for T2 and T1 dimensions. Direct and long-range heteronuclear coupling constants were set to 145 Hz and 8 Hz, respectively. Proton homocorrelated bidimensional experiments (TOCSY) were recorded using 2 K and 256 data point for T2 and T1 dimensions and mixing time was set to 90 ms. Chemical shifts are referred to 7.1 ppm and 77.3 ppm for ^1^H and ^13^C, respectively.

### 5.8. RNA Sequencing, Sequence Assembly and Functional Annotation

RNA sequencing was performed on RNA extracted from AR of *L. austriacum* collected at five experimental points after elicitation with MeJA (control, 2, 5, 24 and 96 h) in three biological replicates. Total RNA were extracted using Nucleozol reagent (Macherey-Nagel, Duren, Germany) following the manufacturer’s instructions. Libraries for Illumina sequencing (a total of 15) were prepared with the TruSeq Stranded RNA Library Prep Kit and produced on average 24 million of 150 bps pair end reads per library (Macrogen, Seoul, Korea). The software BBDuk (sourceforge.net/projects/bbmap/, accessed on 15 December 2020) was used to perform a quality check on the raw sequencing data: minimum length was set to 35 bps and the quality score to 35. The resulting high-quality reads, on average 82.60%, were used as input to perform transcriptome assembly after normalization (with Trinity v2.8.3). Each sample was first normalized individually and then combined in a final normalization step. By using Trinity (v2.8.3), a raw assembly comprising 170,702 transcripts was produced.

To reduce the amount of possible artefacts, three approaches were used. First, the redundancy of the dataset was reduced with the software CD-HIT-EST. Second, the expression levels of all transcripts were quantified with the software Kallisto, and then all the transcripts with no expression levels were removed. Third, only the transcripts with a detectable open reading frame were conserved.

The assembly quality was then assessed with two methods, the first by mapping the normalized reads back to the assembly, and the second by using the BUSCO3 pipeline. As regards the first method, about 85.34% of the reads could be mapped back to the assembly, showing that most of the reads were used during the assembly and it can be considered representative of the biological samples. As regards the BUSCO3 pipeline, two datasets were used, the first comprised 303 proteins that are conserved across all eukaryotes, while the second one contained 430 proteins conserved across plants. The results for the final assembly showed that about 96% of the conserved single copy eukaryotic genes could be found in the assembly.

The putative function of the assembled transcripts was obtained with the TransDecoder pipeline. Open reading frames (>100 bps) of each transcript were extracted, and the best likely coding region per transcript was determined based on a calculated log-likelihood score and a blastp search against *L. usitatissimum* proteome. The final transcriptome assembly was annotated and description was assigned to all the transcripts, although 11,182 transcripts could not be assigned and were classified as “Unknown Protein” (Appendix A set).

The expression quantification of the assembled transcripts was performed with the software Kallisto [82]. Transcripts per Million (TPM) values were calculated for all the transcripts. Similar to the RPKM/FPKM, the TPM procedure normalizes both for the total number of reads and for transcript lengths. The overall quality of the experiment was then evaluated, on the basis of the similarity between replicates, by a Principal Component Analysis (PCA) using the normalized gene expression values as input. Once the consistency of the samples was confirmed by PCA, a differential expression analysis was performed with the package NOISeq [83] to identify the transcripts significantly differentially expressed between samples. The identification of the differentially expressed transcripts was performed with the package NOISeq, the threshold for significance was FDR < 0.05. As each comparison produced several thousands of differentially expressed transcripts, an additional filter was then applied to the fold change (FC), keeping the transcripts with an absolute log_2_FC ≥ 1. Transcriptome assembly and identification of differentially expressed transcripts was performed at the SEQUENTIA BIOTECH SL (Spain).

### 5.9. Fluorescence and Confocal Microscopic Analysis of Justicidin B in Adventitious Root Cultures

Ten single adventitious roots randomly selected at different times after elicitation (1 to 4 weeks) were fixed in 4% (*w*/*v*) paraformaldehyde in phosphate buffer 0.1 M pH 7.4 for 30 min at room temperature. After fixation, roots were washed for 15 min in the same buffer and mounted on a microscope slide with 50% (*v*/*v*) glycerol, then covered with a coverslip sealed with nail polish. In each slide, three different roots (elicited, non-elicited, and negative control for production of justicidin B) were aligned side by side to better compare the degree of fluorescence. Slides were examined by a UV light microscope (Olympus BX 50, Tokyo, Japan) with excitation filter 330–385 nm and barrier filter 420 nm. Some slides were further examined with an inverted confocal laser scanned microscope (CLSM, Nikon A1+, Minato, Japan) equipped with a HeNe laser. Autofluorescence of samples were obtained by exciting at 366 nm and the emission filter was set at 430–490 nm.

### 5.10. Hairy Roots Cultivation in Bioreactor System

Approximately 10 g of FW of HRc line 7 of *L. austriacum* were cultured in 1 L stirred tank bioreactor (Infors HT Labfors, Bottmingen-Basel, Switzerland) containing 650 mL of fresh free-hormones MS medium for 3 and 5 weeks. The vessel was modified with a plastic mesh as root support and separate from the stirring blades. The bioreactor was maintained at 25 °C in the dark, aerated with sterile air at 80 mL/min. The fluid mixing was carried out at 50 rpm. The pH of the medium and the percentage of dissolved oxygen were continuously monitored.

The lignan extraction was performed from fresh tissues dissolved in 80% (*v*/*v*) methanol, subjected to Ika Ultra Turrax T18 and extracted overnight on a rotary shaker at room temperature in the dark. Extracts were then centrifuged for 20 min at 2800× *g* and the clear supernatants was extracted four times with 5 mL chloroform each, the combined organic layers were dried, filtered and rotoevaporated. The recovery and the degree of purity was assessed by HPLC analysis. The extracted roots were dried at 105 °C until constant weight.

## Figures and Tables

**Figure 1 ijms-22-02507-f001:**
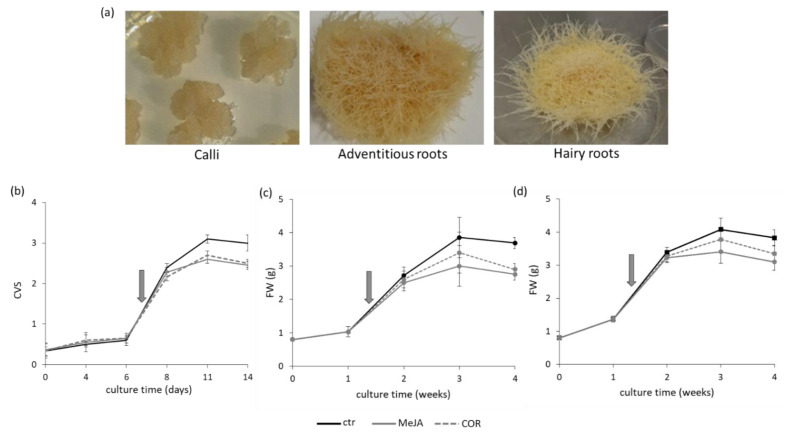
Growth curve of *Linum austriacum* cultures; control (ctr) and treated with methyljasmonate (MeJA) or coronatine COR). (**a**) Pictures of different in vitro cultures obtained from *L. austriacum*. (**b**) Cell cultures (Cc); (**c**) Adventitious root cultures (ARc); (**d**) Hairy root cultures (HRc). CVS: cell volume after sedimentation, FW: fresh weight expressed in grams (g). Arrows indicate the time of elicitation. Each value represents the average of three biological replicates ± SD. Student’s t-test was applied, Cc showed a significance of treated samples versus control at 11 days (*p* ≤ 0.01) and at 14 days (*p* ≤ 0.05); ARc showed a significance of treated samples versus control at 4 weeks (*p* ≤ 0.01); HRc showed a significance of treated versus control at 3 and 4 weeks (*p* ≤ 0.05).

**Figure 2 ijms-22-02507-f002:**
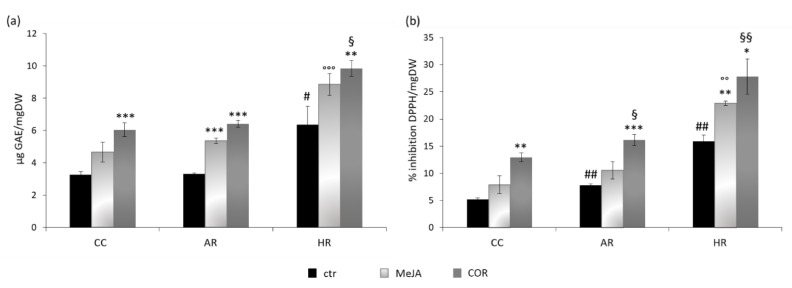
Total phenolic accumulation and radical scavenging activity in *Linum austriacum* cultures. (**a**) Total phenol content in cell (Cc), adventitious root (ARc) and hairy root (HRc) cultures expressed as µg of gallic acid equivalent (GAE) per mg dry weight (DW). (**b**) Radical scavenging activity in Cc, ARc and HRc expressed as percentage of inhibition of 2,2-diphenyl-1-picrylhydrazyl (DPPH) per mg DW. Each bar represents the average of three biological replicates ± SD. Student’s *t*-test was applied: * indicates significance of treated samples versus control per each tissue (* *p* ≤ 0.05; ** *p* ≤ 0.01; *** *p* ≤ 0.001), ^#^ indicates significance among the control samples (ctr) of the different tissues (^#^
*p* ≤ 0.05; ^##^
*p* ≤ 0.01); ° indicates significance among methyljasmonate (MeJA)-treated samples of the different tissues (°° *p* ≤ 0.01; °°° *p* ≤ 0.001); and ^§^ indicates significance among coronatine (COR)-treated samples of the different tissues (^§^
*p* ≤ 0.05; ^§§^
*p* ≤ 0.01).

**Figure 3 ijms-22-02507-f003:**
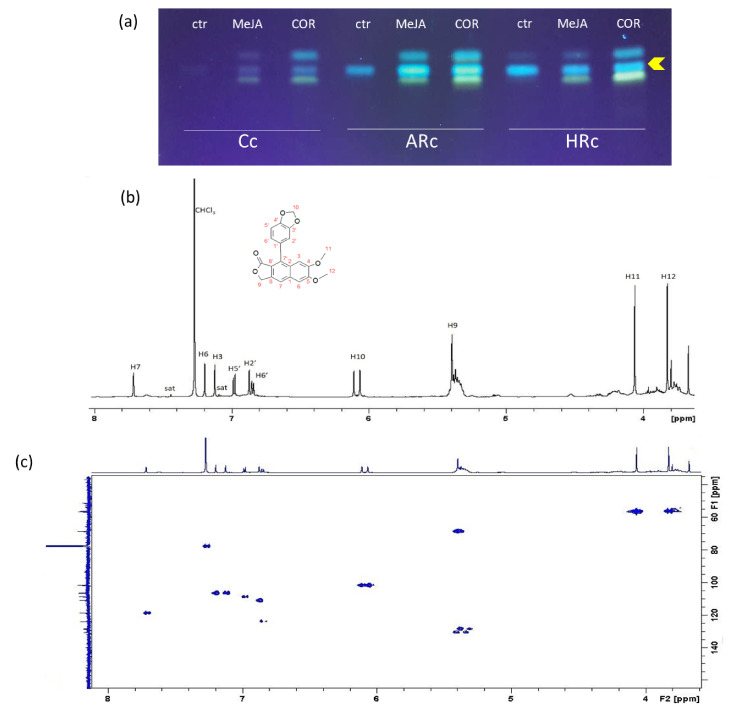
Isolation and identification of justicidin B by TLC and NMR, respectively. (**a**) TLC plate of cell (Cc), adventitious root (ARc) and hairy root (HRc) cultures extracts of control (ctr) and treatments (MeJA and COR) scanned at 366 nm after running in chloroform–methanol 99:1; control (ctr) and treated (MeJA and COR). The arrow indicates the band collected and analyzed by ^1^H-NMR. (**b**), ^1^H-NMR and (**c**) ^1^H-^13^C HSQC spectra of the purified extract. Structure and resonance assignment of justicidin B are reported.

**Figure 4 ijms-22-02507-f004:**
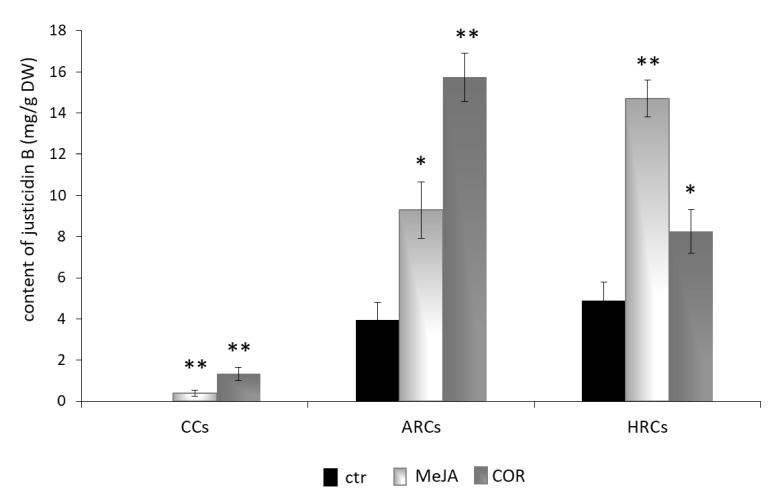
Content of justicidin B in cell (Cc), adventitious root (ARc) and hairy root (HRc) cultures of control (ctr) or treatments (MeJA and COR) expressed as mg per g dry weight (DW); control (ctr) or treated (MeJA and COR). Each bar represents the average of three biological replicates ± SD. Comparisons of differences between the means of the treated and ctr samples were performed using Student’s *t*-tests (* *p* ≤ 0.05; ** *p* ≤ 0.01).

**Figure 5 ijms-22-02507-f005:**
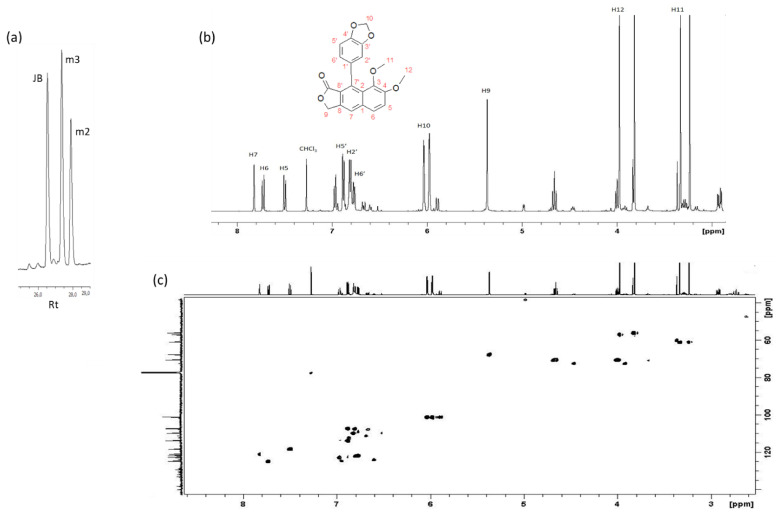
HPLC chromathogram and NMR spectra of *Linum austriacum* extracts. (**a**) Example of high purification liquid chromatography (HPLC) chromatogram from ARc elicited extracts of the justicidin B region (Rt: retention time) showing justicidin B (JB) and other two molecules, m2 and m3. The full chromatogram is reported in Appendix A. (**b**) ^1^H-NMR spectrum of the extracts showing the presence of isojusticidin B (m3). Signals of other molecules in both aliphatic and aromatic regions are present. (**c**) The corresponding ^1^H-^13^C HSQC of the purified extracts, showing the ^1^H-^13^C connectivities. Structure and resonance assignment are reported.

**Figure 6 ijms-22-02507-f006:**
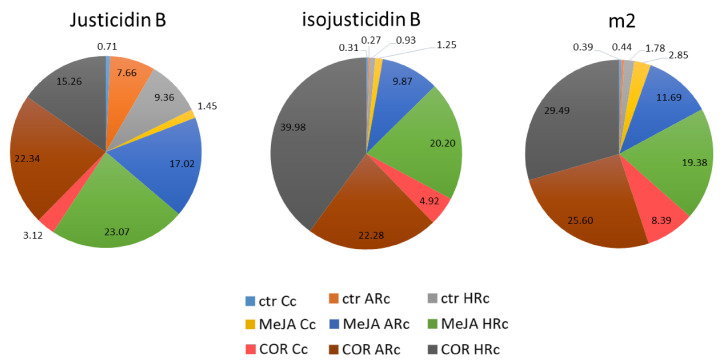
Pie chart with the relative abundance of justicidin B, isojusticidin B and m2 in cell (Cc), adventitious root (ARc) and hairy root (HRc) cultures of control (ctr) and treatments (MeJA and COR). The percentage of each compound per tissue and treatment is reported.

**Figure 7 ijms-22-02507-f007:**
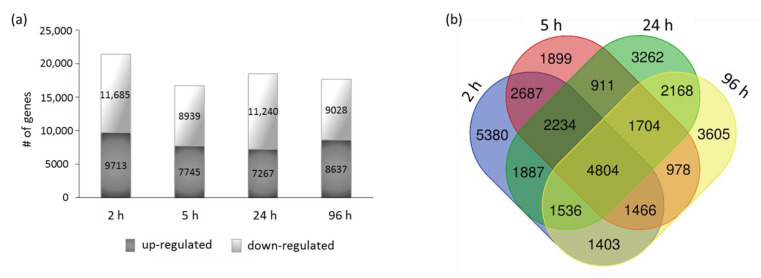
Number of transcripts differentially expressed (DETs) between the control and the elicited samples. (**a**) Total number of DETs between control and treated samples after 2, 5, 24 and 96 h from the elicitation. Numbers within histograms indicate the number of up- and down-regulated DETs. (**b**) The Venn diagram showed the number of DETs shared and unique between different time points.

**Figure 8 ijms-22-02507-f008:**
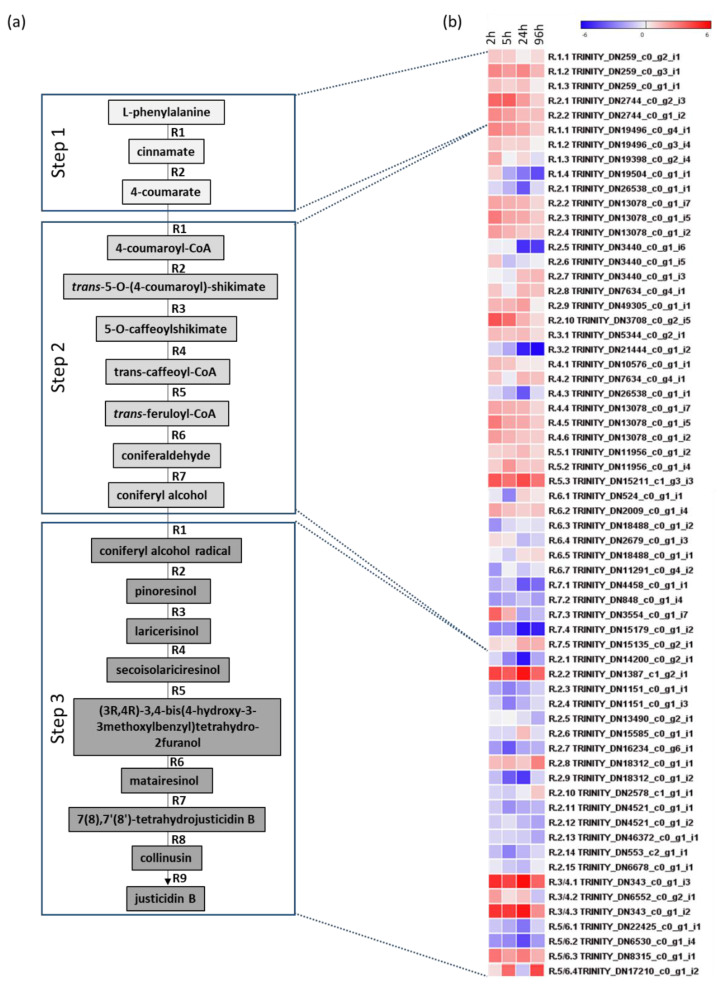
(**a**) A schematic representation of reactions that lead to the formation of justicidin B starting from L-phenylalanine in *Linum austriacum*. The pathway is divided for simplicity in three main steps (step 1, step 2 and step 3) with their respective reactions (R). (**b**) Heat map of differentially expressed transcripts (DETs) identified for each reaction of the three steps.

**Figure 9 ijms-22-02507-f009:**
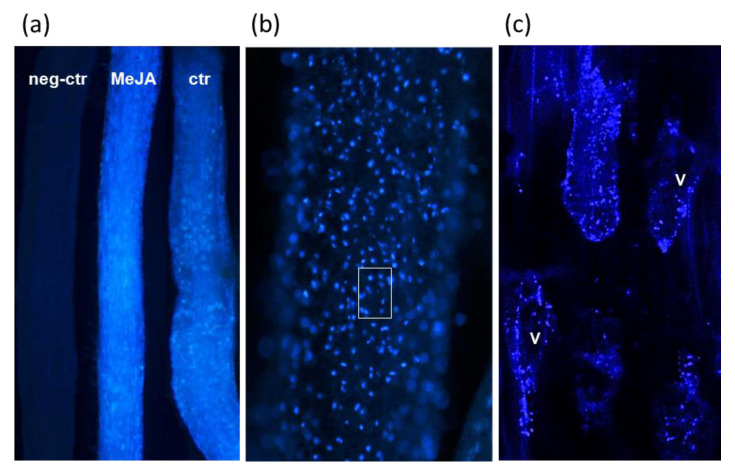
Microscopic analysis of justicidin B localization in adventitious roots of *Linum austriacum*. (**a**) Autofluorescence of justidicin B in adventitious root culture (ARc) of comparable age of *L. austriacum* and *L. mucronatum.* Neg-ctr: ARc of *L. mucronatum*, a justicidin B no producing *Linum* species (negative control) and MeJA: -elicited ARc of *L. austriacum*, ctr: not-elicited ARc of *L. austriacum*. The three randomly chosen roots were mounted on the same slide for a clear comparison of fluorescence intensity, which is slightly higher in MeJA in comparison with ctr and absent in neg-ctr. (**b**) Enlargement of a MeJA-treated root in which very brilliant fluorescence spots appear, apparently located inside cortical parenchyma cells. (**c**) The cortical tissue framed in (**b**) when scanned with confocal laser scanning microscope (CLSM) at 366 nm reveals that the spots are in turn formed by much smaller spots mainly located on cell periphery, thus likely in the cytoplasm, but not on the wall or in the vacuole (V).

**Figure 10 ijms-22-02507-f010:**
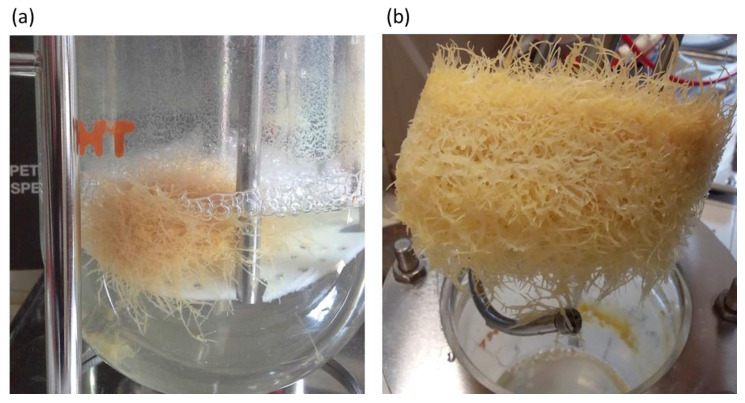
*Linum austriacum* HRc in bioreactor. (**a**) Mesh root-supported airlift reactor containing *L. austriacum* HRc7 line culture. (**b**) HRc7 line culture collection after 21 days of growth.

## Data Availability

Not applicable.

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
