# Peer review of "New Insight into Justicidin B Pathway and Production in Linum austriacum"

_ijms, 2021, doi:10.3390/ijms22052507_

Round 1

Reviewer 1 Report

The paper describes the enhanced production of Justicidin B by hairy roots from L. austriacum, followed by an important study on the mechanism of elicitation of the production by some precursors.

It is an important paper that should lead the way towards the description and understanding of the intimate mechanisms of the enzymatic pathways leading to natural compounds, beside the particular result concerning the production of Justicidin B in hairy roots.

Another naïve question: Would it be possible to derive cells from the roots and make them produce the compound?

It is clearly written and is a nice demonstration. A few mostly minor points should be fixed. As suggested in the following sections:

Important: Please, the differences between the curves of figures 1 a, b and c are small. I wonder what a significance statistical test would look like. Not sure they are different (especially in the HRc), even though a tendency is seen.

Minor: Lines 52-4: Is the section Linum different from the genus Linum??? Or is it a typo?

                Line 69-70: Interesting statement… Does anything more specific is known about this loop mechanism? [This was written as I read the paper. I later understood that it is one of the core points of the paper. I (strongly) believe that should be more emphasized in the introduction.]

                Figure 1: Shouldn’t the arrows be positioned vertically instead of horizontally, because they point out at a time point?

Line 189: The Authors should mention in 1 sentence from what the analyses were performed? Extracts (as mentioned in the legend of Figure 5)? (if yes, ethanolic, etc…)??? (even if it is later mentioned in the exp section).

More importantly, in Figure 5, a full chromatogram 366 and 210 nm (??) should be shown, not the portion of it (as shown in Figure 5A). And the wavelength of the detection should be mentioned either on the figure itself (y-axis) or in the corresponding legend.

Figure 6: Although pie charts are clearly didactic – and nice colors!! – I wonder if 3 sets of histograms would not be clearer…?

Lines 227 & 243: no comma after “although”

Lines 230-231: please, change “15% (5380, 2 hours), 5.30% (1899, 5 hours), 9.10% (3262, 24 hours) and 10% (3605, 96 hours)” for “15% (5380 at 2 hours), 5.30% (1899 at 5 hours), 9.10% (3262 at 24 hours) and 10% (3605 at 96 hours)”. For some readers “5380,2 hours” can be read “5380.2 hours”. This is misleading – although perfectly correct.

Line 247: “…no information was available for the … ” should be put at the present tense: “…no information is available for the…”

Line 299-300: Naïve question: The control Hr from L. mucronatum does not produce Justicidin B, correct? Maybe repeat this there, just for the general reader??

Finally, I find the discussion a bit long and somewhat disorganized. Maybe a focus on the main results could be performed? Again, this is a suggestion as if the Authors want to make sure their message-to-take-home is understood and taken home…

Author Response

RESPONSES to REFEREE 1:

Thank you very much to the referee for the important suggestion that we addressed below.

Important:

Point 1: Please, the differences between the curves of figures 1 a, b and c are small. I wonder what a significance statistical test would look like. Not sure they are different (especially in the HRc), even though a tendency is seen.

Response 1: Following the referee suggestion a Student’s t-test has been applied to the Figure 1 data and the results have been reported in Figure 1 legend. Cell cultures showed a significance of treated samples versus control at 11 days (p ≤ 0.01) and at 14 days (p ≤ 0.05); ARc showed a significance of treated samples versus control at 4 weeks (p ≤ 0.01); HRc showed a significance of treated versus control at 3 and 4 weeks (p ≤ 0.05).

Minor:

Point 2: Lines 52-4: Is the section Linum different from the genus Linum??? Or is it a typo?

Response 2: The section is a secondary taxonomic rank positioned between genus and species. As reported by Schmidt et al 2012 the genus Linum is divided into 5-6 sections : Syllinum, Cathartolinum and Linopsis, Linum and Dasylinum. The last two sections mainly accumulate as major lignans arylnaphthalene-(AN) type such as justicidin B (Schmidt et al., 2006, 2010).  Lines 52-54 have been modified to better explain this taxonomic difference.

Point 3: Line 69-70: Interesting statement… Does anything more specific is known about this loop mechanism? [This was written as I read the paper. I later understood that it is one of the core points of the paper. I (strongly) believe that should be more emphasized in the introduction.]

Response 3: We are really sorry but we do not understand which is the “loop mechanism” that the reviewer refers to. Lines 69-70 are the following: “Although the production of secondary metabolites is often low (less than 1% DW) [24], it could be improved by elicitation with substances that initiates or enhances the biosynthesis of a specific compound when introduced in small quantities in the living cell system”.

Point 4: Figure 1: Shouldn’t the arrows be positioned vertically instead of horizontally, because they point out at a time point?  

Response 4: Figure 1 has been revised and the arrows positioned vertically.

Point 5: Line 189: The Authors should mention in 1 sentence from what the analyses were performed? Extracts (as mentioned in the legend of Figure 5)? (if yes, ethanolic, etc…)??? (even if it is later mentioned in the exp section). 

Response 5: the text corresponding has been modified mentioning which extracts have been used for the analyses: “Therefore, the methanolic ARc elicited extract was subjected to a further purification to fully remove the justicidin B…”.

Point 6: in Figure 5, a full chromatogram 366 and 210 nm (??) should be shown, not the portion of it (as shown in Figure 5A). And the wavelength of the detection should be mentioned either on the figure itself (y-axis) or in the corresponding legend.

Response 6: The full HPLC chromatogram has been reported as supplementary figure S3 and the wavelength of detection has been reported in the legend of the figure.

Point 7: Figure 6: Although pie charts are clearly didactic – and nice colors!! – I wonder if 3 sets of histograms would not be clearer…?

Response 7: we have modified the pie chart to better clarify the results obtained indicating on the pie chart the relative percentage of each molecule in each tissue per each treatment. The addition of these values on the chart clarify the relative amount of each molecule.

Point 8: Lines 227 & 243: no comma after “although” 

Response 8: In lines 227 and 243 the comma after “although” has been deleted.

Point 9: Lines 230-231: please, change “15% (5380, 2 hours), 5.30% (1899, 5 hours), 9.10% (3262, 24 hours) and 10% (3605, 96 hours)” for “15% (5380 at 2 hours), 5.30% (1899 at 5 hours), 9.10% (3262 at 24 hours) and 10% (3605 at 96 hours)”. For some readers “5380,2 hours” can be read “5380.2 hours”. This is misleading – although perfectly correct.

Response 9: in lines 230-231 the changes suggested have been made, i.e. “15% (5380 at 2 hours)….”

Point 10: Line 247: “…no information was available for the … ” should be put at the present tense: “…no information is available for the…”

Response 10: line 247 has been modified  using the present tense: “… no information is available for the…”

Point 11: Line 299-300: Naïve question: The control Hr from L. mucronatum does not produce Justicidin B, correct? Maybe repeat this there, just for the general reader?? 

Response 11:  a sentence with the explanation regarding the use of L. mucronatum as negative control has been added in the text: “L. mucronatum was chosen as negative control since it belongs to a Linum section producing aryltetralin-type lignans [11,14].”  

Point 12: Finally, I find the discussion a bit long and somewhat disorganized. Maybe a focus on the main results could be performed? Again, this is a suggestion as if the Authors want to make sure their message-to-take-home is understood and taken home…

Response 12: to help the reader follow the discussion, this section has been divided into paragraphs with the relative subheadings. Moreover, a “Conclusion” chapter has been added to focus on the main results of the paper.

Reviewer 2 Report

The authors of this study focused on the in vitro production of important secondary plant metabolites (lignans) by Linum austriacum which are valuable both for plant life and human health. In particular, the authors studied cells (Cc), adventitious roots (ARc), and hairy roots (HRc) for the production of justicidin B through elicitation with methyl jasmonate (MeJA) and coronatine (COR). Using cutting-edge technologies and methodologies as well as a robust experimental design, the authors obtained rich results and evaluated them properly. They have managed to identify and quantify the targeted metabolites; they have provided a comprehensive overview of the targeted pathway and the reactions leading to the formation of the targeted metabolite. In addition, they have identified in silico the activated genes involved. Furthermore, this study reveals for the first time the intracellular localization of justicidin B in ARc through microscopic analysis. Last but not least, the authors attempted to scale-up their findings using a bioreactor. Undoubtedly, the results presented in this study provide new insight related to the justicidin B pathway and its cellular localization in L. austriacum and may channel new research initiatives in the future.

The abstract is well-written, providing an overview of the aim, methodologies, and results. There are just a few linguistic imperfections that can be easily corrected.

The introduction is targeted, concise, and not overwhelming, ending up with the well-defined aims of the study.

Material and methods are quite detailed and fully presented. The only basic mistake that can be avoided is to avoid using ‘minutes’ and prefer min instead. All these minor changes that are necessary are indicated with sticker notes in the reviewed pdf file.

The results section is quite rich, novel findings are clearly presented and these are well-illustrated with original figures, charts, chromatograms, spectra, and images. Only Figure 6 suffers somehow from imprecision. The authors are invited to indicate the values for the labels in Figure 6. Otherwise, they should refer to these values in the manuscript (lines 211-215).

Discussion is targeted to aims without repetitions of results. The authors discuss their findings with regard to contemporary and relevant scientific literature. The authors’ attempt to discuss the justicidin B pathway is solid and concise. Therefore, I can hardly make any suggestions for improvement. The discussion part is not short and its length (three pages) is adequate and well-balanced with other sections. However, it is not structured and therefore is not very easy to follow. Perhaps the manuscript could benefit from a more structured discussion section with some subheadings.

The supplementary materials presented are quite rich and indispensable. However, their legends are quite short and probably need improvement to be fully understood in a glance of an eye. If the legends are written more carefully and in detail, their readership and usefulness would be improved.

References are well-formatted and fully given with all associated details. There only some imperfections that can be easily corrected by italicizing the scientific names of some plants. All these minor changes that are necessary are indicated with sticker notes in the pdf file reviewed (3-14 stickers per page).

Last but not least, I have to point out that there are several linguistic imperfections detected throughout the manuscript. I have tried to suggest rephrasing or better syntax where appropriate in order to facilitate understanding and allow flowing. I think that the manuscript could benefit a lot from these suggestions. All these are indicated with sticker notes in the reviewed pdf file (3-14 stickers per page).

Author Response

RESPONSES to REFEREE 2

Thank you very much to the referee for the important suggestions that we addressed below.

Moreover, all the corrections indicated in pdf file have been revised following the referee suggestions.

Point 1: Material and methods are quite detailed and fully presented. The only basic mistake that can be avoided is to avoid using ‘minutes’ and prefer min instead. All these minor changes that are necessary are indicated with sticker notes in the reviewed pdf file.

Response 1: the authors checked the whole paper and corrected minutes with the abbreviation min.

Point 2: The results section is quite rich, novel findings are clearly presented and these are well-illustrated with original figures, charts, chromatograms, spectra, and images. Only Figure 6 suffers somehow from imprecision. The authors are invited to indicate the values for the labels in Figure 6. Otherwise, they should refer to these values in the manuscript (lines 211-215).

Response 2: the authors indicated on Figure 6 the values corresponding to each slice of the pie chart.

Point 3: Discussion is targeted to aims without repetitions of results. The authors discuss their findings with regard to contemporary and relevant scientific literature. The authors’ attempt to discuss the justicidin B pathway is solid and concise. Therefore, I can hardly make any suggestions for improvement. The discussion part is not short and its length (three pages) is adequate and well-balanced with other sections. However, it is not structured and therefore is not very easy to follow. Perhaps the manuscript could benefit from a more structured discussion section with some subheadings.

Response 3: The authors provided a new discussion section divided in paragraphs identified by subheadings. Moreover, a new section was added: “Conclusion” reporting the main novelties of the manuscript.

Point 4: The supplementary materials presented are quite rich and indispensable. However, their legends are quite short and probably need improvement to be fully understood in a glance of an eye. If the legends are written more carefully and in detail, their readership and usefulness would be improved.

Response 4: the authors provided new legends more detailed.

Point 5: References are well-formatted and fully given with all associated details. There only some imperfections that can be easily corrected by italicizing the scientific names of some plants. All these minor changes that are necessary are indicated with sticker notes in the pdf file reviewed (3-14 stickers per page).

Response 5: the authors corrected the scientific names of the species not italicized.

Point 6: Last but not least, I have to point out that there are several linguistic imperfections detected throughout the manuscript. I have tried to suggest rephrasing or better syntax where appropriate in order to facilitate understanding and allow flowing. I think that the manuscript could benefit a lot from these suggestions. All these are indicated with sticker notes in the reviewed pdf file (3-14 stickers per page).

Response 6: the authors checked all the corrections indicated by the stickers in the pdf file.
